# Multi-Class Procedure for Analysis of 50 Antibacterial Compounds in Eggshells Using Ultra-High-Performance Liquid Chromatography–Tandem Mass Spectrometry

**DOI:** 10.3390/molecules26051373

**Published:** 2021-03-04

**Authors:** Małgorzata Gbylik-Sikorska, Anna Gajda, Ewelina Nowacka-Kozak, Beata Łebkowska-Wieruszewska, Andrzej Posyniak

**Affiliations:** 1Department of Pharmacology and Toxicology, National Veterinary Research Institute (NVRI), al. Partyzantow 57, 24-100 Pulawy, Poland; anna.gajda@piwet.pulawy.pl (A.G.); ewelina.nowacka@piwet.pulawy.pl (E.N.-K.) aposyn@piwet.pulawy.pl (A.P.); 2Department of Pharmacology, Toxicology and Environmental Protection, University of Life Sciences, 20-950 Lublin, Poland; lebkowska.wieruszewska@up.lublin.pl

**Keywords:** eggshell, eggs, laying hens, UHPLC–MS/MS, antimicrobial, multi-class

## Abstract

In this work, for the first time, Ultra-High-Performance Liquid Chromatography–Tandem Mass Spectrometry (UHPLC–MS/MS) method was developed for qualitative and quantitative analysis of veterinary antibiotics (cephalosporins, diaminopyrimidines, fluoro(quinolones), lincosamides, macrolides, penicillins, pleuromutilins, sulfonamides, tetracyclines, and sulfones) in hen eggshells. The sample preparation method is based on a liquid–liquid extraction with a mixture of metaphosphoric acid, ascorbic acid, EDTA disodium salt dihydrate, and acetonitrile. The chromatographic separation was performed on Luna^®^ Omega Polar C18 10 column in gradient elution mode and quantitated in an 8 min run. Validation such as linearity, selectivity, precision, recovery, matrix effect, limit of quantification (LOQ), and limit of detection (LOD) was found to be within the acceptance criteria of the validation guidelines of the Commission Decision 2002/657/EC and EUR 28099 EN. Average recoveries ranged from 81–120%. The calculated LOQ values ranged from 1 to 10 µg/kg, the LOD values ranged from 0.3 to 4.0 µg/kg, depending on analyte. The developed method has been successfully applied to the determination of antibacterial compounds in hen eggshell samples obtained from different sources. The results revealed that enrofloxacin, lincomycin, doxycycline, and oxytetracycline were detected in hen eggshell samples.

## 1. Introduction

Hen eggshells make up about 10% of the total egg weight [1,2,3], which means that egg producers and the egg-using industry generate a huge amount of waste. The U.S. Environmental Protection Agency (EPA), ranked eggshell waste as the 15th major food industry pollution issue [4,5]. European Commission regulations also indicate that eggshell waste could be considered hazardous [5,6]. The European Union is the second-largest egg producer in the world, the first being China. Egg production in the EU increased by about half a million tonnes between 2010 and 2018 [7]. Annual egg production in the EU and the U.K. between 2010 and 2018 ranged from 6272 to 6940 (×1000 tonnes). In 2019 and 2020, it is estimated that production was around 7065 and 7144 (×1000 tonnes), respectively [8]. *The future of food and agriculture–Alternative pathways to 2050 report*, presents potential scenarios for global egg production in 2030 and 2050. The “towards sustainability scenario” (TSS) represents a proactive shift towards more sustainable food and farming systems in which global egg production is predicted to reach 85 million tons in 2030 and 91 million tons in 2050 [9]. The possibility of making use of such a large amount of biowaste is a very big challenge and an important element of environmental protection. 

Due to their unique composition and chemical structure, hen eggshells can be used in many industries. They consist of a mineralized shell (94% calcium carbonate, 1% magnesium carbonate, 1% calcium phosphate, and trace elements) and a fibrous structure, the eggshell membrane [2,10,11,12]. Eggshell waste has found applications in the biofuel industry. Due to its high calcium carbonate content, they can be used as a heterogeneous catalyst in the transesterification process [4,12,13]. They can also be used to adsorb aquatic environment pollutants (toxic heavy metal, organic compounds, hydrogen sulfide, and dyes) [4,13,14,15,16]. Some researchers have found a way to use eggshells as a carbon dioxide sorbent in CCS (carbon capture and storage) [17]. Other authors have also reported the possibility of applying eggshells as bio-filters for polypropylene composites [18]. Another way to reuse biowaste such as eggshells is to use them as crop fertilizer. They provide a very good natural source of calcium, which helps some plants grow in calcium-deficient soil, and they balance the soil pH [3,4,13,19,20,21]. Eggshells are a natural biomaterial, so they have found practical applications in various fields of medicine. They can be a source of calcium for the production of hydroxyapatite or nan-calcium citrate which can be used in orthopedics as a bone graft substitute [4,6,11,13]. In another application, they can be used as a nutritional additive in animal feed [3,4,13]. Moreover, because of their mineral content, especially calcium in a very easily bioavailable form, they can be used in the pharmaceutical industry to produce commercially available calcium supplements [4,10,13,22], and in the food industry as a source of dietary calcium in fortified food [23,24] or beverages [25]. In the available literature, we also found information about the preparation of eggshell powder at home and the selection of the correct amount for a natural calcium supplement [26]. 

There are four main systems of raising laying hens in the EU: in enriched cages (50.4%), in barns (27.8%), free-range (16.3), or organic (5.4) [27]. Large-scale animal husbandry results in the accumulation of numerous animals in a relatively small area, which contributes to the more rapid spread of bacterial diseases. Despite the application of the principles of biosecurity aimed at improving animal welfare and increasing animal resistance to possible infections, the use of antibiotics is often the only solution and the only effective way to reduce losses resulting from the spread of diseases. The use of veterinary drugs (antibiotics) in hens during the laying period is strictly limited. Some of these compounds are absorbed by the digestive tract of laying hens and transferred to the eggs [28]. To protect consumer health, the EU established maximum residue limits (MRLs) in eggs for only a few antibiotics (chlortetracycline, oxytetracycline, tetracycline, erythromycin, lincomycin, neomycin, tiamulin, tylosin, and colistin) [29]. The list of antibiotics approved by the Food and Drug Administration (FDA) for laying hens in the United States is also severely limited: bacitracin, erythromycin, hygromycin B, and tylosin [30]. Despite numerous restrictions and bans on the use of antibiotics, they are still sometimes used inappropriately or illegally in animal husbandry. Reports of the European Food Safety Authority (EFSA) from 2010 to 2018 showed that the number of non-compliant results for antibacterials (B1) in eggs were found in 93 samples [31,32,33,34,35,36,37,38,39]. The most frequently detected drugs were enrofloxacin (*n* = 34) and doxycycline (*n* = 24) as well as different sulfonamides (*n* = 20), all of which are prohibited in the EU for use in hens during the laying period. From 2015 to 2020, five cases of antibiotic detection in eggs and egg products were registered in the Rapid Alert System for Food and Feed (RASFF), of which enrofloxacin was the most frequently detected compound even at a concentration of 4236 µg/kg (in 2020). The presence of antibiotic residues in food and animal products such as eggs can lead to adverse effects, the most common of which are allergic reactions. Some antibiotics have the potential to cause immunopathological effects, nephropathy, hepatotoxicity, reproductive disorders, and even mutagenicity or carcinogenicity [40,41,42,43]. The most important adverse effect of antibiotic residues in food is the possibility of inducing antimicrobial resistance (AMR), which has become a significant global threat to human and animal health [44]. Global consumption of antimicrobials in food animal production was estimated at 63,151 t in 2010 and is projected to rise by 67%, to 105,596 t by 2030. The Centers for Disease Control and Prevention (CDC) indicated that about 2,000,000 people were infected with antibiotic resistant bacteria, resulting in 23,000 deaths in the U.S. in 2013 [45]. If nothing changes, by 2050 up to 10,000,000 people per year could lose their lives [44].

The possibility of identifying a new source of antibiotic residues is an important element in preventing and eliminating antibiotic-resistant bacteria in the environment. This approach is in agreement with the “One Health” concept, which aims to strengthen collaboration in various sectors, such as public health, animal health, plant health, and the environment. Antibiotics used in human and veterinary medicine are made of the same or very similar molecules, and it can be expected that there will be a transfer of resistance between humans and animals either directly or via the environment. Therefore, detailed knowledge of new sources of antibiotic residues that could induce antimicrobial resistance in humans, animals, and the environment is of great importance for human health.

There are many papers in the literature on drug residues in eggs and their distribution between the yolk and egg white [28,46,47,48,49], but there are no studies on the possible transfer of drugs into the hen eggshells. The possibility of using eggshells as a food additive, supplement, animal feed additive, or plant fertilizer may risk exposing humans, animals, and the environment to antibiotic residues. Therefore, it is very important to study the possibility of transferring antibiotics into the eggshells. 

This study presents a multi-class UHPLC–MS/MS analytical method for the qualitative and quantitative analysis of different commonly used veterinary antibiotics (cephalosporins, diaminopyrimidines, fluoro(quinolones), lincosamides, macrolides, penicillins, pleuromutilins, sulfonamides, tetracyclines, and sulfones) in hen eggshells. The developed method was validated according to international requirements: linearity, selectivity, specificity, precision (repeatability and within-laboratory reproducibility), and recovery. In addition, the limit of quantification (LOQ) and limit of detection (LOD) were estimated. This method has been successfully applied to the determination of antibiotic residues in hen eggshell samples obtained from different sources.

## 2. Results and Discussion

### 2.1. Optimization of LC–MS/MS Conditions

The UHPLC–MS/MS method for determining 50 analytes in eggshells was investigated. The multiple reaction monitoring (MRM) fragmentations were optimized for each analyte by direct infusion of analyte standard solution into the mass spectrometer. For quantitation and confirmation, generally, the protonated parent ions [M + H]^+^ and two transition products were monitored (Table 1). The mass source parameters such as declustering potential (DP), collision energy (CE), and dwell time were optimized in positive ionization mode separately for each analyte. 

The LC separation was optimized for all studied compounds. Different mobile phases: water; 0.075%, 0.1%, 0.5% formic acid (water phase) with acetonitrile; methanol; mixture of acetonitrile with methanol (80:20, 50:50, 20:80 *v*/*v*); 0.05%, 0.075%, 0.1% formic acid in acetonitrile; and 0.05%, 0.075%; 0.1% formic acid in methanol (organic phase), using three different LC columns: Agilent ZORBAX SB-C18 column (50 mm × 2.1 mm, 1.8 µm); a Phenomenex Luna^®^ Omega Polar C18 10 column (100 mm × 2.1 mm, 1.6 µm) and Phenomenex Luna C18 (2) 100A column (50 mm × 3.0, 3 µm) were tested. Also, various gradient programs with different flow rates were tested to obtain optimal separation for all compounds in a relatively short runtime. The most efficient separation, best peak shape along with the highest possible signal intensity, was successfully achieved with the mobile phase consisted of 0.075% formic acid and 0.05% formic acid in acetonitrile combined with Luna^®^ Omega Polar C18 10 column. Chromatograms of eggshell samples spiked with a mixture of 50 compounds are presented in Figure 1.

### 2.2. Optimization of Sample Preparation

In the first step, the sample preparation procedure was optimized. The avian eggshell is composed of about 96% calcium carbonate in the form of calcite and other non-organic components with an organic matrix (3.5%), comprising the eggshell membranes and other constituents [1,2,3]. A very important stage was to improve the way of cleaning the eggshell of possible contamination and cross-contamination, and three different ways were tested: water, water then methanol, and methanol then water. The best results were obtained with the use of methanol followed by ultrapure water. This cleaning method also improved the separation of the shell from the membrane. Various ways of grinding eggshells were then tested: milling in the ball mixer mill; grinding from being frozen in liquid nitrogen, and blending. The best results were obtained from using the ball mixer mill, which ground the eggshells into a powder of similar grain size. When using the ball mill, it was necessary to select the appropriate parameters (grinding time, frequency, type, and ball size). The most efficient grinding was achieved by using a combination of 4 stainless steel balls of different sizes: 2 balls Ø 10 mm and 2 balls Ø 20 mm. Three grams of hand-crushed eggshell were weighted into a 35 mL stainless steel grinding jar with the addition of the balls, and milling was carried at a frequency of 25Hz at room temperature for 3 min. Using the method of grinding eggshells frozen in liquid nitrogen, we obtained similar results, but this method was more expensive and time-consuming. 

Different extraction procedures were then investigated to obtain the best analyte isolation from a specific matrix such as eggshells. So far in the available literature, there are no publications determining the antibacterial drug residues left in eggshells. Taking into account our previous experience in developing multi-class methods in different food matrices, we decided to test a different mixture of acids with organic solvents in the extraction step. The following extraction mixtures were tested: formic acid; metaphosphoric acid; ascorbic acid; citric acid; acetic acid (different pH and concentration) with acetonitrile and methanol. Initially, the results showed that the use of 1% metaphosphoric acid pH = 5.0 in combination with 0.5% ascorbic acid pH = 4.0 and acetonitrile showed the best recoveries and optimal results for all compounds. The use of formic acid, acetic acid, and citric acid in combination with acetonitrile or methanol did not give satisfactory results; in most cases, the extraction was not efficient. The use of a mixture of metaphosphoric acid with acetonitrile made it possible to isolate analytes from the matrix, but the addition of ascorbic acid improved the extraction of groups such as sulfonamides and fluoro(quinolones). Moreover, some analytes (tetracyclines, fluoro(quinolones)) can form chelates with metal ions (e.g., Ca ^2+^, calcium carbonate), which is the primary component of the eggshell and contains about 40% Ca ^2+^ ions [50]. So, the addition of EDTA disodium salt dihydrate was tested to improve the recovery of these analytes, the recoveries for tetracyclines were significantly higher. Figure 2 summarizes the comparison of the mean recoveries between the different classes of compounds for the 5 selected best results achieved for the different extraction mixtures tested.

The extraction efficiency was enhanced by testing the use of an ultrasonic bath and a rotary stirrer at different time intervals (15 min, 30 min, 45 min, 1 h, 1.5 h). The best results were obtained with a 45 min ultrasonic bath. After extraction and purification, the recovery of all analyzed compounds was determined in this study.

Additionally, to reduce possible interfering components in the final extract, two different syringe filter membranes (PTFE and PVDF) were tested. Finally, the use of PVDF filters allows for the best results to be achieved for all analytes without recovery reduction. 

### 2.3. Method Validation

All of the matrix-matched calibration curves were linear (determination coefficient, r^2^), over the range of LOQ–1000 µg/kg, and were above 0.998 for all compounds. The within-laboratory reproducibility and repeatability were satisfactory for each analyte. The coefficients of variation (CVs, %) for repeatability was lower than 10% and 15% for within-laboratory reproducibility, and the results are shown in Table 2. The specificity and selectivity of the method were verified by analyzing blank eggshell samples, which allowed us to verify that no peaks from endogenous compounds were detected in the retention time corresponding to each analyte or internal standard (IS). The average recoveries were in the range of 81–120%. The LOQ and LOD values determined for the developed method are shown in Table 2, the calculated LOQs were in the range of 1 to 10 µg/kg, the LODs were in the range of 0.3 to 4.0 µg/kg. 

The matrix effects (MEs) were within the acceptable limits (81–128%) after evaluating a mixture of 5 different lots of eggshells. The ME, % value between 85 to 115% was considered as “not to be observed”. The obtained results indicate that the ME was observed in 26% of analytes (Table 2). Ion suppression was observed in most cases, ion enhancement was observed for some sulfonamides and OXA. To minimize and avoid the matrix effect, it is recommended that matrix-match calibration curves be used in each analysis of these substances in eggshell samples.

### 2.4. Application of Real Samples

To evaluated the applicability of the method, the eggshell samples (*n* = 20) were collected from eggs obtained from several different sources: 10 eggs were taken from two different experiments carried out on laying hens that received multiple oral doses of enrofloxacin (*n* = 3) or lincomycin (*n* = 3), and the eggs were collected on the 5th day of the administration of the antimicrobial drug. Other egg samples were obtained from two experiments in which doxycycline (*n* = 2) or oxytetracycline (*n* = 2) depletion and residues in eggs were determined after a single oral administration of the drug, and the eggs were collected 2 and 4 days after the end of the doxycycline and oxytetracycline administration, respectively. Eggshell samples (*n* = 10) obtained from eggs taken in 2020 as part of the National Residue Control Plan (NCP) for surveillance of veterinary drug residues in food of animal origin were also analyzed. The results of eggshell samples analysis obtained from experiments in which the drug was administered to animals proved that substances such as enrofloxacin, lincomycin, doxycycline, and oxytetracycline are distributed and present in the eggshell. The individual results are summarized in Table 3. This also confirms the need to control the antibacterial compound residues in eggshells, especially when they can be used in the food industry or as a calcium supplement. Selected sample chromatograms are shown in Figure 3. 

## 3. Materials and Methods

### 3.1. Chemical and Reagents

Acetonitrile and methanol, LC–MS-grade, were obtained from J.T. Baker (Deventer, The Netherlands), EDTA disodium salt dihydrate and sodium hydroxide were purchased from POCH, Gliwice, Poland; ascorbic acid, metaphosphoric acid and formic acid (≥ 95% for LC–MS) came from Sigma–Aldrich (St. Louis, MO, USA). Syringe filters PVDF (0.22 µm) were purchased from Restek (Bellefonte, PA, USA). Ultrapure water, the conductivity of at least 18 MΩ/cm was prepared with the Millipore purification system.

The cefquinome (CFQ), cefalonium (CFLO), cefazolin (CFZ), cephalexin (CFLE), cefoperazone (CFPE), cefapirin (CFPI), ceftiofur (CFT), trimethoprim (TMP), trimethoprim-d9 (TMP-d9), ciprofloxacin (CIP), enrofloxacin (ENR), danofloxacin (DAN), difloxacin (DIF), flumequine (FLU), marbofloxacin (MAR), sarafloxacin (SAR), norfloxacin (NOR), oxolinic acid (OXO), nalidixic acid (NAL), ciprofloxacin-d8 (CIP-d8) lincomycin (LIN), josamycin (JOS) erythromycin (ERY), spiramycin (SPI), tylosin (TYL), tulathromycin (TLM), tilmicosin (TIL), azytromycin (AZY) amoxicillin (AMOX), penicillin G (PEN G), penicillin V (PEN V), ampicillin (AMPI), dicloxacillin (DICLOX), cloxacillin (CLOX), nafcillin (NAF), oxacillin (OXA), penicillin G-d7 (PEN G-d7), piperacillin (PIP), valnemulin (VAL), tiamulin (TIM), sulfamerazine (SME), sulfamethazine (SMT), sulfadimethoxine (SDMX), sulfamethoxazole (SMA), sulfamonomethoxine (SMM), sulfathiazole (SFT), sulfaquinoxaline (SQX), sulfadoxine (SDX), sulphamethoxypyridazine (SMP), sulfadiazine (SDZ), sulfafenazole (SFF), doxycycline (DC), tetracycline (TC), oxytetracycline (OTC), chlortetracycline (CTC), demeclocycline (DMC), and dapson (DDS) were purchased from Sigma–Aldrich (St. Louis, MO, USA). 

### 3.2. Preparation of the Standard Stock Solutions and Working Solutions

Stock standard solutions (1000 µg/mL) of cephalosporins and penicillins were prepared in ultrapure water and stored in polypropylene vessels. Diaminopyrimidines, lincosamides, macrolides, pleuromutilins, sulfonamides, sulfones, and tetracyclines were dissolved in methanol, fluoro(quinolones) were dissolved in methanol with the addition of 1M sodium hydroxide (99:1, *v*/*v*), and were stored in glass vessels. All individual stock standard solutions were stable for at least 6 months when retained in a dark place at −18 °C. A mixture of working solutions (1 µg/mL) and mixture of internal standard (IS) solutions was obtained by dilution in ultrapure water and stored at 4–8 °C for 1 month. 

### 3.3. Sample Preparation

In the sample pretreatment step, eggshells were first cleaned with methanol and ultrapure water to remove contaminants; then, the eggshell membrane was removed to reduce possible cross-contamination. The eggshells thus prepared were dried at room temperature for 1 h, transferred to string bags and hand crushed. Three grams of crushed eggshell were weighed in a stainless steel 35 mL grinding jar with the addition of 4 stainless steel, 2 balls Ø 10 mm and 2 balls Ø 20 mm. Milling was carried out in the ball mixer mill MM 400 (Retsch–Verder Scientific, Haan, Germany) at a frequency of 25 Hz at room temperature for 3 min. Such milled eggshells (2 g) were weighed into 10 mL centrifuge tubes; then, 40 µL of IS mixture (1 µg/mL) was added, mixed, and stored in a dark place at 4–8 °C for 15 min. After incubation, 750 µL of 1% metaphosphoric acid pH = 5.0, 500 µL 0.5% ascorbic acid pH = 4.0, 200 µL 0.1M EDTA disodium salt dihydrate, and 8 mL of acetonitrile were added to the eggshells. The samples were vortexed for 30 s and put in an ultrasonic bath (Bandelin SONOREX™, Berlin, Germany) for 45 min at room temperature. Then, the samples were centrifuged at 2930 × rcf for 10 min, temperature: 4 °C (Centrifuge 4-16KS, Sigma, Darmstadt, Germany). The supernatants were evaporated to dryness at 45 ± 5 °C under a stream of nitrogen, redissolved in 500 µL of ultrapure water, and filtered by syringe filters (0.22 µm PVDF) into LC vials prior to LC–MS/MS analysis.

### 3.4. LC–MS/MS Analysis

The Shimadzu Nexera X2 (Shimadzu, Kyoto, Japan) ultra-high-performance liquid chromatograph (UHPLC) system coupled to the QTRAP^®^ 4500 triple-quadrupole mass spectrometer (Sciex, Framingham, MA, USA) was used for sample analysis. 

The UHPLC system was equipped with a Luna^®^ Omega 1.6 µm Polar C18 10 column (100 mm × 2.1 mm, Phenomenex, Torrance, CA, USA) integrated with a guard column of the same type at 35 °C. The mobile phase consisted of two eluents, 0.075% formic acid (A) and 0.05% formic acid in acetonitrile (B). The samples were separated under the following gradient conditions: 92% A (0.01–4.00 min), 90% B (4.01–6.30 min), and 92% A (6.31–8.00 min), at a flow rate of 0.32 mL/min. A volume of 3 µL of sample extract was injected. The mass spectrometric analysis was performed using triple-quadrupole detection in positive ion mode (ESI+). The instrument was set to collect data in multiple reaction monitoring mode (MRM) for quantification. The ion transitions and mass parameters monitored for each analyte are listed in Table 1. The following MS/MS parameters were used for multi-compound analysis: source temperature = 470 °C; IonSpray voltage = 5500 V; Curtain Gas = 20 psi; ion source gas 1 = 40 psi; ion source gas 2 = 50 psi; Entrance potential = 10 V.

### 3.5. Method Validation

The method was validated according to the Commission Decision 2002/657/EC [51]. Linearity, selectivity, specificity, precision (repeatability and within-laboratory reproducibility), and recovery were evaluated. In addition, the limit of quantification (LOQ) and limit of detection (LOD) were estimated (EUR 28,099 EN) [52]. Linearity in matrix (antibiotics free eggshells) was tested in the range LOQ-1000 µg/kg at 7 concentration levels: LOQ (1.0, 5.0, 10.0, depending on the analyte); 20, 50, 100, 200, 500, and 1000 µg/kg. Selectivity and specificity were investigated by analysis of 6 different blank eggshell samples to test for potential interference with endogenous substances. Method repeatability was estimated after analyzing 6 samples spiked at 3 concentration levels: 10, 50, and 100 µg/kg by the same operator on the same day with the same instrument, after which CVs (%) were calculated. The within-laboratory reproducibility was calculated as an overall CV (%) of the results, which obtained after an analysis of fortifying another two sets of blank samples with the same concentration levels. It was repeated on two different days with the same instrument and different operators. The average recovery was evaluated in the same experiment as repeatability by comparing the mean measured concentration with the fortified concentration of the samples. The LOD was determined at the signal-to-noise ratio (S/N) ≈ 3; the LOQ was defined as the lowest validated concentration with the S/N > 10. The matrix effect (ME, %) was assessed comparing matrix-matched standards (the signal intensity of a sample extract fortified after extraction–*IM*) with standards in solvent (the signal intensity of fortified water–*IW*) at the corresponding concentration of 50 µg/kg, which is expressed by the following Equation (1):
(1)$$\mathrm{ME}(\%)=\frac{IM}{IW}\times 100\;$$


## 4. Conclusions

The possibility of identifying a new source of antibiotic residues is one element in preventing and eliminating antimicrobial resistance in the environment. A first attempt was made to develop a multi-class UHPLC–MS/MS method for determining 50 different antimicrobial compounds in hen eggshells to carry out relevant studies. The presented method has been successfully validated in accordance with Commission Decision 2002/657/EC and guidelines EUR 28,099 EN. It was found that this analytical method can be successfully used in the analysis of antibacterial residues in hen eggshells. Moreover, according to our knowledge, it is the first time that antibacterial residues have been detected in eggshell samples. Studies conducted on real eggshell samples may provide evidence for the possibility of drug transfer into the eggshells. Conducting further research into the potential for the distribution of various veterinary drugs into eggshells is of great importance to human health.

## Figures and Tables

**Figure 1 molecules-26-01373-f001:**
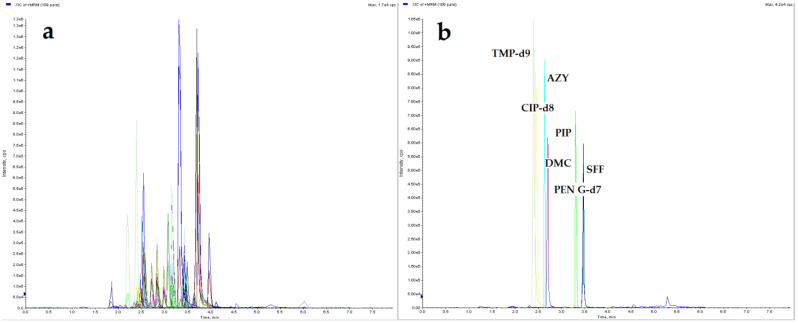
UHPLC–MS/MS XICs, (**a**) eggshell sample spiked with 50 analytes at 10 µg/kg and internal standard (IS) mixture at 20 µg/kg; (**b**) blank eggshell sample with IS mixture at 20 µg/kg.

**Figure 2 molecules-26-01373-f002:**
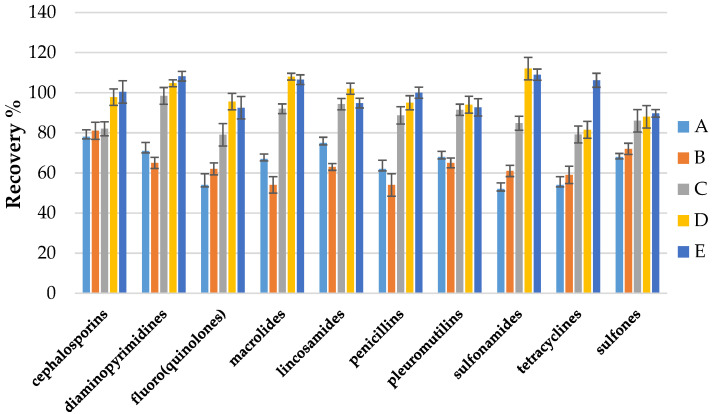
Effect of different extraction condition on recovery of 10 antibacterial compounds classes: (**A**) formic acid+ acetonitrile; (**B**) acetic acid + methanol; (**C**) metaphosphoric acid + acetonitrile; (**D**) metaphosphoric acid + ascorbic acid + acetonitrile; (**E**) metaphosphoric acid + ascorbic acid + EDTA + acetonitrile.

**Figure 3 molecules-26-01373-f003:**
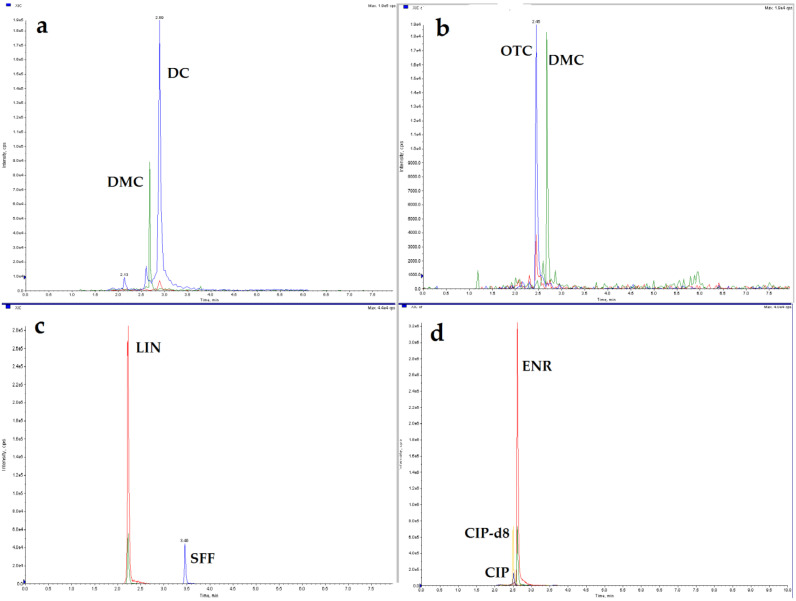
Chromatograms of real eggshell samples: (**a**) doxycycline 17.5 µg/kg; (**b**) oxytetracycline 26.4 µg/kg; (**c**) lincomycin 12.0 µg/kg; (**d**) enrofloxacin 587 µg/kg; and ciprofloxacin 18.3 µg/kg.

**Table 1 molecules-26-01373-t001:** Summary of the MRM for analytes and MS/MS parameters.

Group	Analyte	Ion Transition 1 (*m*/*z*)	Ion Transition 2 (*m*/*z*)	Retention Time (min)	DP (V)	CE * (eV)
cephalosporins	CFQ	529.0/134.0	529.0/125.0	2.28	50	25
	CFLO	459.0/337.0	459.0/152.0	2.43	46	16
	CFZ	455.0/323.0	455.0/156.0	2.71	50	15
	CFLE	348.0/158.0	348.0/106.0	2.36	50	10
	CFPE	646.0/530.0	646.0/530.0	2.93	50	35
	CFPI	424.0/152.0	424.0/124.0	2.16	50	35
	CFT	524.0/241.0	524.0/125.0	3.16	50	25
diaminopyrimidines	TMP	292.0/262.0	292.0/231.0	2.42	52	36
	TMP-d9(IS)	300.0/234.0	-	2.39	55	34
fluoro(quinolones)	CIP	332.0/314.0	332.0/231.0	2.48	65	28
	ENR	360.0/342.0	360.0/286.0	2.59	100	33
	DIF	400.0/382.0	400.0/356.0	2.75	50	30
	DAN	358.0/340.0	358.0/255.0	2.53	60	33
	FLU	262.0/244.0	262.0/202.0	3.75	44	25
	MAR	363.0/345.0	363.0/320.0	2.42	70	30
	SAR	385.0/368.1	385.0/348.0	2.72	50	31
	NOR	320.0/302.0	320.0/231.0	2.45	50	30
	OXO	262.0/244.0	262.0/216.0	3.33	53	25
	NAL	233.0/215.0	233.0/187.0	3.69	42	30
	CIP-d8(IS)	340.0/322.0	-	2.46	60	29
lincosamides	LIN	407.0/126.0	407.2/359.0	2.21	74	36
macrolides	ERY	734.0/576.0	734.0/158.0	3.13	75	28
	TYL	916.0/174.0	916.0/772.0	3.22	110	51
	TLM	806.6/577.4	806.6/230.0	2.30	61	33
	TIL	869.0/696.0	869.0/174.0	2.83	135	56
	JOS	828.0/173.0	828.0/229.0	2.63	80	46
	SPI	843.0/540.0	843.5/174.0	2.61	120	44
	AZY(IS)	749.0/591.0	-	2.63	89	40
penicillins	AMOX	366.0/349.0	366.0/208.0	2.04	45	12
	PEN G	335.0/160.0	335.0/176.0	3.51	60	17
	PEN V	351.0/160.0	351.0/114.0	3.66	55	16
	AMPI	350.0/106.0	350.0/160.0	2.34	58	27
	DICLOX	470.0/160.0	470.0/311.0	2.75	50	20
	CLOX	436.0/160.0	436.0/277.0	3.92	40	18
	NAF	415.0/199.0	415.0/171.0	3.97	50	20
	OXA	402.0/160.0	402.0/243.0	3.78	50	18
	PEN G-d7(IS)	342.0/183.0	-	3.46	35	20
	PIP(IS)	540.0/398.0	-	3.31	55	24
pleuromutilins	VAL	565.0/263.0	565.0/156.0	3.49	45	40
	TIM	494.0/192.0	494.0/118.0	3.43	128	30
sulfonamides	SME	265.0/156.0	265.0/108.0	2.72	40	25
	SMT	279.0/156.0	279.0/108.0	3.20	50	25
	SDMX	311.0/156.0	311.0/108.0	3.17	50	23
	SMA	254.0/107.0	254.0/155.0	3.20	42	24
	SMM	281.0/156.0	281.0/108.0	3.00	50	35
	SFT	256.0/156.0	256.0/108.0	2.57	53	20
	SQX	301.0/156.0	301.0/108.0	1.85	50	23
	SDX	310.9/156.0	310.9/108.0	3.45	60	25
	SMP	280.0/156.0	280.0/108.0	2.84	60	25
	SDZ	251.0/156.0	251.0/108.0	2.55	53	22
	SFF(IS)	315.0/156.0	-	3.46	90	26
tetracyclines	OTC	461.0/426.0	461.0/444.0	2.45	50	28
	TC	445.0/410.0	445.0/427.0	2.57	55	27
	CTC	479.0/444.0	479.0/462.0	2.81	50	28
	DC	445.0/428.0	445.0/154.0	2.87	60	23
	DMC(IS)	465.0/448.0	-	2.69	60	17
sulfones	DDS	248.9/156.0	248.9/108	3.08	50	19

* The CE value for ion transit 1.

**Table 2 molecules-26-01373-t002:** Validation results.

Analyte	Repeatability *, (CV,%)	Within-Lab Reproducibility *, (CV,%)	LOQ (µg/kg)	LOD (µg/kg)	Recovery * (%)	Matrix Effect * (%)
CFQ	8.8 ± 1.0	14.1 ± 1.0	5.0	1.0	119.1 ± 3.6	112.3 ±1.2
CFLO	9.1 ± 1.1	12.8 ± 0.5	5.0	1.0	106.4 ± 5.4	95.4 ± 2.0
CFZ	7.9 ± 1.0	14.5 ± 1.2	5.0	1.0	81.6 ± 3.2	89.2 ± 2.3
CFLE	6.4 ± 1.1	10.4 ± 0.7	5.0	1.0	83.1 ± 3.5	102.3 ± 1.1
CFPE	6.8 ± 1.0	10.1 ± 1.0	5.0	1.0	97.7 ± 4.1	96.7 ± 1.2
CFPI	7.1 ± 1.1	13.7 ± 0.8	5.0	1.0	103.1 ± 3.5	94.4 ± 1.4
CFT	7.9 ± 1.0	12.1 ± 1.1	5.0	1.0	107.7 ± 4.1	119.7 ± 1.3
TMP	8.2 ± 1.1	12.2 ± 0.6	5.0	1.0	92.2 ± 5.5	112.0 ± 1.0
CIP	9.9 ± 1.0	13.1 ± 1.1	1.0	0.3	82.8 ± 6.1	125.3 ± 2.0
ENR	7.1 ± 1.1	10.7 ± 0.8	1.0	0.3	105.7 ± 3.6	101.5 ± 2.2
DIF	6.6 ± 1.0	10.1 ± 1.3	1.0	0.3	118.3 ± 4.1	88.6 ± 2.1
DAN	7.8 ± 1.1	11.6 ± 0.8	1.0	0.3	102.8 ± 3.6	107.6 ± 1.6
FLU	7.9 ± 1.0	12.1 ± 1.4	1.0	0.3	84.9 ± 3.1	107.1 ± 2.2
MAR	7.3 ± 1.1	11.4 ± 0.4	1.0	0.4	104.5 ± 5.8	102.5 ± 2.1
SAR	7.0 ± 1.0	11.1 ± 1.1	1.0	0.3	108.9 ± 2.3	89.4 ± 1.4
NOR	8.9 ± 1.1	12.3 ± 0.7	1.0	0.4	97.1 ± 5.6	94.3 ± 1.0
LIN	6.8 ± 1.0	12.1 ± 1.0	1.0	0.4	116.2 ± 3.6	96.7 ± 1.3
ERY	6.7 ± 1.1	12.5 ± 0.9	1.0	0.4	105.5 ± 5.6	101.3 ± 1.6
TYL	6.4 ± 1.0	12.1 ± 1.1	5.0	1.0	97.7 ± 3.7	124.7 ± 1.1
TIL	6.4 ± 1.1	12.8 ± 0.9	5.0	1.0	99.5 ± 4.2	122.7 ± 1.8
JOS	8.8 ± 1.0	13.1 ± 0.6	1.0	0.4	105.2 ± 4.4	104.4 ± 0.9
SPI	8.4 ± 1.0	13.9 ± 0.7	5.0	1.0	116.0 ± 3.9	116.4 ± 2.1
TLM	8.8 ± 0.5	14.6 ± 0.4	10.0	4.0	115.8 ± 4.6	111.4 ± 1.7
AMOX	8.9 ± 0.3	11.2 ± 0.4	1.0	0.4	85.1 ± 4.6	89.7 ± 1.3
PEN G	7.6 ± 1.2	11.4 ± 0.9	1.0	0.3	89.3 ± 3.8	103.1 ± 0.8
PEN V	7.9 ± 1.0	10.1 ± 1.1	1.0	0.5	108.3 ± 5.1	116.3 ± 0.6
AMPI	6.3 ± 1.1	10.6 ± 0.7	1.0	0.5	114.3 ± 4.7	127.3 ± 1.4
DICLOX	7.9 ± 1.0	11.1 ± 1.3	1.0	.03	105.2 ± 2.7	125.2 ± 1.1
CLOX	9.1 ± 1.1	13.5 ± 0.4	1.0	0.5	86.4 ± 4.6	127.4 ± 1.4
NAF	8.5 ± 1.0	11.1 ± 1.0	1.0	0.3	98.4 ± 3.6	91.3 ± 2.1
OXA	8.0 ± 1.1	12.8 ± 0.9	1.0	0.3	93.3 ± 2.1	84.6 ± 0.5
TIM	6.9 ± 1.0	11.1 ± 0.4	1.0	0.3	106.8 ± 3.6	111.2 ± 1.2
VAL	6.5 ± 1.1	10.7 ± 0.6	1.0	0.4	96.9 ± 3.1	106.1 ± 1.6
SMT	7.3 ± 1.0	11.1 ± 1.1	1.0	0.3	94.5 ± 3.8	106.6 ± 1.1
SME	7.6 ± 1.1	11.8 ± 0.5	1.0	0.3	108.9 ± 2.3	118.7 ± 1.3
SDMX	8.1 ± 1.0	12.1 ± 1.3	1.0	0.3	112.1 ± 3.6	97.6 ± 0.9
SMA	7.2 ± 1.1	12.7 ± 0.9	1.0	0.3	111.6 ± 5.6	108.2 ± 1.7
SMM	7.3 ± 0.5	12.1 ± 0.4	1.0	0.3	115.1 ± 3.2	97.6 ± 2.1
SFT	8.7 ± 1.1	11.4 ± 0.9	1.0	0.4	85.8 ± 3.8	104.8 ± 2.3
SQX	8.6 ± 1.0	13.5 ± 1.1	1.0	0.3	89.3 ± 4.6	81.9 ± 0.8
SDX	7.1 ± 1.1	12.6 ± 0.7	1.0	0.3	85.7 ± 4.9	83.8 ± 1.4
SMP	8.4 ± 1.0	11.9 ± 0.7	1.0	0.3	108.2 ± 4.3	98.2 ± 1.0
SDZ	8.8 ± 0.5	10.6 ± 0.4	1.0	0.3	118.6 ± 3.4	97.9 ± 0.8
DC	8.9 ± 0.3	10.2 ± 0.4	1.0	0.4	102.0 ± 4.9	88.5 ± 1.0
OTC	7.6 ± 1.2	12.4 ± 0.6	1.0	0.4	91.4 ± 3.6	103.8 ± 0.8
TC	6.4 ± 1.1	10.8 ± 0.8	1.0	0.4	93.3 ± 2.8	98.3 ± 0.5
CTC	9.8 ± 1.0	13.1 ± 1.1	1.0	0.4	105.4 ± 4.1	95.9 ± 1.2
DDS	6.6 ± 1.1	10.1 ± 0.9	1.0	0.4	85.5 ± 3.6	87.4 ± 0.3

* average of 3 validation levels with standard deviation ( ± SD).

**Table 3 molecules-26-01373-t003:** Results of eggshell samples analysis.

Sample	Analyte	Concentration (µg/kg)	Rout/Dose (mg/kg bw *)/Time of Treatment (Days)
1 (experiment 1)	ENR CIP	587 18.3	oral/10/5
2 (experiment 1)	ENR CIP	423 10.5	oral/10/5
3 (experiment 1)	ENR CIP	634 12.4	oral/10/5
4 (experiment 2)	LIN	12.0	oral/20/5
5 (experiment 2)	LIN	9.6	oral/20/5
6 (experiment 2)	LIN	15.4	oral/20/5
7 (experiment 3)	DC	17.5	oral/10/1
8 (experiment 3)	DC	14.6	oral/10/1
9 (experiment 4)	OTC	26.4	oral/20/1
10 (experiment 4)	OTC	18.7	oral/20/1
11 (NRCP)	nd	-	-
12 (NRCP)	nd	-	-
13 (NRCP)	nd	-	-
14 (NRCP)	nd	-	-
15 (NRCP)	nd	-	-
16 (NRCP)	nd	-	-
17 (NRCP)	nd	-	-
18 (NRCP)	nd	-	-
19 (NRCP)	nd	-	-
20 (NRCP)	nd	-	-

* bw = body weight.

## Data Availability

The data presented in this study are available in the manuscript.

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
