# Peer review of "Multi-Class Procedure for Analysis of 50 Antibacterial Compounds in Eggshells Using Ultra-High-Performance Liquid Chromatography–Tandem Mass Spectrometry"

_molecules, 2021, doi:10.3390/molecules26051373_

Round 1

Reviewer 1 Report

The article is written in a very concise and clear way, easy to read and understand. Overall they highlighted the importance of their work in preventing and eliminating the phenomenon of antimicrobial resistance and provided a robust and ready-to-use method that could be further used as a practical application.

In the article written by Gbylik-Sikorska et al., 2021, the authors elaborate an extraction method and detection method for 50 Antibacterial Compounds in Eggshells by means of Ultra-High-Performance Liquid Chromatography-Tandem Mass Spectrometry.

The paper seems to be a continuation of the previous paper published by Anna Gajda of the article authored by TOMASZ BŁĄDEK et al., 2012, entitled - MULTI-CLASS PROCEDURE FOR ANALYSIS OF ANTIBACTERIAL COMPOUNDS IN EGGS BY LIQUID CHROMATOGRAPHY-TANDEM MASS SPECTROMETRY. In the previous work, they indeed evaluated the fresh eggs but using different procedures.  

The authors did the qualitative and quantitative analysis of different commonly used veterinary antibiotics  like cephalosporins, diaminopyrimidines, fluoro(quinolones), lincosamides, macrolides, penicillins, pleuromutilins, sulfonamides, tetracyclines, and sulfones) in hen eggshells. The developed method was validated according to international requirements: linearity, selectivity, specificity, precision (repeatability and within-laboratory reproducibility), and recovery. The limit of quantification (LOQ) and limit of detection (LOD) were also estimated. 

The actual paper is indeed new and important (the search on PubMed with the keywords “antibiotic-resistant AND eggshell” – retrieved 5 results, “LC-MS AND eggshell”- 2 results, “antibiotics AND eggshell”- 123  results). This study is aiming to identify and raise awareness on the important sources of antibiotic residues like the eggshell, contributing to the prevention and elimination of antibiotic-resistant bacteria phenomenon in the environment.

However, a few minor revision can be added in order to further improve the quality of the paper:

  1.       The exact method of calculation of the parameters regarding method validation (linearity, LODs, and LOQs, accuracy, and precision), including formulas that were used can be added.
  2.       If possible, a representative figure summarizing the different effects of the extraction condition on analytes recoveries (maybe a representative per class of compounds) can be added.
  3.        A new section should be added to the materials and methods section. This section should describe in detail the animal experiments. Details regarding the exact quantity of administrated antibiotics should be included. Also, in table nr 3, the addition of a new column including the type of treatment and administrated concentration of drug should be included to offer an overview regarding the amount of antibiotic that is detected in the eggshell comparative with the administrated amount.

Reviewer 2 Report

Comments to authors:

After a careful reading of the manuscript, the topic of your manuscript is interesting. It is well organized, well written and easy to understand. Most of references are recent, appropriate and closely related to the previous work. All these points are advantages of this work.

I have some comments below.

  1. Results and Discussion

Table1: CE was inserted for only one transition, although there are two transitions in the table, please specify which transition 1, or 2 corresponding to the inserted CE or better to insert the CE for each transition?

2.2. Optimization of Sample Preparation

-crushed eggshell were weighted into a 35ml stainless steel grinding jar with the addition of balls, milling was carried at a frequency of 25Hz at room temperature for 3 minutes….. there is space 35 ml, 25 Hz.

2.3. Method Validation

- 120%. The LOQ and LOD values determined for the developed method end are shown ….  Not end, should be and.

- Table 2. Validation results

 The average recoveries percents for each spiking concentration (10; 50 and 100 μg/kg) should be inserted in the table?

- HPLC-MS/MS chromatogram of a standard mixture of 50 compounds plotted as overlapped multiple reaction monitoring (MRM) transition of each analyte, should be inserted?

  1. Materials and Methods

3.2. Preparation of the Standard Stock Solution and Working Solutions

All stock standard solutions were stored in a dark place at – 18 °C for 6 months.... why stored for 6 months (long period)?

3.3. Sample Preparation

The samples were vortexed for 30 sec and put on an ultrasonic bath for 45 min at room temperature…… You should insert model and company of the ultrasonic bath instrument and other devices used in this work!
